# Frequency of Circulating Tumor Cells (CTC) in Patients with Brain Metastases: Implications as a Risk Assessment Marker in Oligo-Metastatic Disease

**DOI:** 10.3390/cancers10120527

**Published:** 2018-12-19

**Authors:** Annkathrin Hanssen, Carlotta Riebensahm, Malte Mohme, Simon A. Joosse, Janna-Lisa Velthaus, Lars Arne Berger, Christian Bernreuther, Markus Glatzel, Sonja Loges, Katrin Lamszus, Manfred Westphal, Sabine Riethdorf, Klaus Pantel, Harriet Wikman

**Affiliations:** 1Department of Tumor Biology, University Medical Centre Hamburg-Eppendorf, 20246 Hamburg, Germany; annkathrin.hanssen@yahoo.de (A.H.); carlotta.riebensahm@insel.ch (C.R.); s.joosse@uke.de (S.A.J.); s.loges@uke.de (S.L.); s.riethdorf@uke.de (S.R.); pantel@uke.de (K.P.); 2Department of Neurosurgery University Medical Centre Hamburg-Eppendorf, 20246 Hamburg, Germany; m.mohme@uke.de (M.M.); lamszus@uke.uni-hamburg.de (K.L.); westphal@uke.de (M.W.); 3Department of Internal Medicine II and Clinic (Oncology Centre) University Medical Centre Hamburg-Eppendorf, 20246 Hamburg, Germany; j.velthaus@uke.de (J.-L.V.); l.berger@uke.de (L.A.B.); 4Institute of Neuropathology University Medical Centre Hamburg-Eppendorf, 20246 Hamburg, Germany; c.bernreuther@uke.de (C.B.); m.glatzel@uke.de (M.G.)

**Keywords:** lung cancer, brain metastasis, circulating tumor cell (CTC), survival, oligo-metastasis, EpCAM

## Abstract

Forty percent of non-small cell lung cancer (NSCLC) patients develop brain metastases, resulting in a dismal prognosis. However, patients in an oligo-metastatic brain disease setting seem to have better outcomes. Here, we investigate the possibility of using circulating tumor cells (CTCs) as biomarkers to differentiate oligo-metastatic patients for better risk assessment. Using the CellSearch^®^ system, few CTCs were detected among NSCLC patients with brain metastases (*n* = 52, 12.5% ≥ two and 8.9% ≥ five CTC/7.5 mL blood) and especially oligo-metastatic brain patients (*n* = 34, 5.9%, and 2.9%). Still, thresholds of both ≥ two and ≥ five CTCs were independent prognostic indicators for shorter overall survival time among all of the NSCLC patients (*n* = 90, two CTC ≥ HR: 1.629, *p* = 0.024, 95% CI: 1.137–6.465 and five CTC ≥ HR: 2.846, *p* = 0.0304, CI: 1.104–7.339), as well as among patients with brain metastases (two CTC ≥ HR: 4.694, *p* = 0.004, CI: 1.650–13.354, and five CTC ≥ HR: 4.963, *p* = 0.003, CI: 1.752–14.061). Also, oligo-brain NSCLC metastatic patients with CTCs had a very poor prognosis (*p* = 0.019). Similarly, in other tumor entities, only 9.6% of patients with brain metastases (*n* = 52) had detectable CTCs. Our data indicate that although patients with brain metastases more seldom harbor CTCs, they are still predictive for overall survival, and CTCs might be a useful biomarker to identify oligo-metastatic NSCLC patients who might benefit from a more intense therapy.

## 1. Introduction

The formation of metastasis is the limiting factor of survival for most carcinoma patients. Despite some improvements in treatment options, mortality among non-small cell lung cancer (NSCLC) still remains extremely high. Forty percent of non-small cell lung cancer (NSCLC) patients already present with metastases at initial diagnosis (stage IV), resulting in a median overall survival of only four months in men and five months in women [1,2]. Metastases hereby originate from single tumor cells, i.e., circulating tumor cells (CTCs), which were initially able to detach from the primary tumor, survive in the circulation, and colonize distant target tissues [3,4]. CTC detection in the peripheral blood of NSCLC patients and patients suffering from other tumor entities can be used for prognosis estimation, and to predict time to metastatic relapse and overall survival [5,6,7,8]. Furthermore, sequential CTC analysis allows the monitoring of treatment success with very limited intervention [9,10]. Thus, CTCs are currently discussed as a promising liquid biopsy marker.

In NSCLC, tumors metastasize most frequently to brain (~40%) and bone (~35%), followed by liver (20%), lungs (20%), and the adrenal glands (8–10%) as target organs [2,11]. Different organs provide unique environmental conditions for tumor cells. In today’s literature, the brain is specifically described as a privileged site, mainly due to the existence of the blood–brain barrier, which prevents most therapeutic agents from entering brain tissue. Brain metastases are associated with a dismal prognosis of only four weeks of overall survival if left untreated [12]. In general, patients with less distant metastatic sites have a better survival outcome compared to patients with a multi-level metastatic disease, suggesting the potential importance of differentiating between oligo-metastatic and multi-metastatic patients [13,14,15]. Lung cancer patients with an oligo-metastatic brain disease (and a controlled primary tumor) significantly benefit from a metastatic resection combined with radiation therapy compared to radiation therapy alone, with a five-year overall survival of up to 21% [16,17,18]. Furthermore, for radically-treated patients, less brain recurrences and an increased performance status were recorded [16,17,19,20,21]. It is noteworthy that the recently published randomized, controlled phase II study demonstrated markedly improved progression-free survival for oligo-metastatic patients receiving local consolidative therapy compared to maintenance therapy [22]. Thus, the differentiation between multi-metastatic and oligo-metastatic NSCLC diseases should be made. The discovery of potent biomarkers to stratify patients with cerebral metastasis into high-risk and low-risk groups potentially enables the identification of patients who might benefit from aggressive therapy and extended adjuvant treatment regimens. Given the increasing efficacy of systemic targeted therapies and the rising numbers of long-term survivors in many tumor entities, it is crucial to identify patients at risk and therapeutic windows to effectively induce tumor remission. CTCs as peripheral biomarkers can potentially identify patients with a higher risk for systemic disease progression.

Here, we analyzed CTC frequencies in peripheral blood from patients with advanced NSCLC metastasized to the brain or other distant sites. To evaluate the prognostic impact of CTCs, we particularly distinguished between an oligo-brain disease (patients with brain as the single metastatic site) and a systemic disease setting. We could demonstrate that patients with brain metastases originating from NSCLC and also other tumor entities more seldom harbor CTCs compared to other metastatic sites. Nevertheless, these CTCs are an independent prognostic factor for overall survival in NSCLC patients, and thus could be used as a biomarker to identify those oligo-brain metastatic NSCLC patients with a true good prognosis. Therefore, they would profit from aggressive treatment, including the resection of both brain metastasis and primary tumor.

## 2. Results

### 2.1. NSCLC Patient Characteristics

The peripheral blood from 90 patients with metastatic NSCLC was analyzed for the presence of EpCAM-positive CTC by CellSearch^®^ analysis. The number and sites of metastases were recorded for each patient at the time of blood collection. A total of 34 patients had an oligo-metastatic brain disease, whereas 22 patients had additional metastases at the time of CTC analysis. Fourteen patients had adrenal gland metastases, 23 had lung metastases beyond the continuous spread of the primary tumor, and 16 had bone metastases. The median survival for all of the patients was 8.7 months (range: 0.3–47.7 months). Detailed patient characteristics are displayed in Table 1.

### 2.2. CTC Detection in Advanced NSCLC Patients

Patients were defined as CTC-positive when keratin-positive but CD45-negative nucleated cells were detected. A cut-off of ≥two CTCs per 7.5 ml of blood identified 13.3% (12/90) of advanced NSCLC patients as CTC-positive. The median CTC count was three (range: two to 19 cells). A cut-off of ≥five per 7.5 mL of blood CTCs revealed 7.8% CTC positivity (median seven; range five to 19).

The fourth CellSearch channel was used to analyze EGFR expression of CTCs in CTC-positive NSCLC patients. Here, 33% of patients (five out of 15) were found to have EGFR-expressing CTCs, with a great heterogeneity in intensity between and within the patients. The EGFR expression ranged from weak (+1) to strong (+3) (Table 2 and Figure 1C). All of the EGFR-positive CTCs were detected in patients with an EGFR wild-type primary tumor, indicating a strong heterogeneity of EGFR expression also between the primary tumor and CTCs. The EGFR expression pattern did not correlate with clinical data.

### 2.3. Clinical Value of CTCs in NSCLC

The prevalence of CTC positivity of NSCLC patients was analyzed in correlation to clinical characteristics. No association of CTC positivity was found with gender, tumor histology, or TNM status (Table 1). CTCs (≥two) were detected more often among patients with adrenal gland metastases (42.9%) compared to other patients (*p* = 0.003, G-test of independence with Williams’ correction). More CTCs were also found among patients with lung (26.1%) metastases compared to patients with brain metastases (12.5%) (Table 1). Only two out of 32 oligo-brain (brain as the only metastatic site) patients (5.9%, ≥two) were found to be CTC-positive, whereas more patients with multiple metastatic lesions were CTC-positive (22.7% (5/22); *p* = 0.063, Figure 1A). No association was found between the number of brain metastases and the number of CTCs (*p* = 0.552, zero-inflated negative binomial regression). We also differentiated between NSCLC patients who have undergone brain metastasis resection (*n* = 34) and patients who had not been operated on their metastatic lesion in the central nervous system (*n* = 23). Herein, CTC incidence was significantly lower in patients who underwent a metastatic resection (2/35, 5.4%), than in patients without operation (5/14, 26.3%, *p* = 0.011, Figure 1B).

With a cut-off of ≥five CTC/7.5 mL blood, 8.9% of brain metastatic patients had CTCs, whereas adrenal and bone metastatic patients had 21.4% and 18.8% CTCs, respectively (Table 1). Only one of the patients operated from brain metastases (2.4%) was detected as CTC-positive (*p* = 0.014; G-test of independence with Williams’ correction).

### 2.4. Overall Survival of NSCLC Can Be Predicted by CTC Analysis

Survival data was available for 87 NSCLC patients, and overall survival analysis was performed for both a CTC cut-off of ≥two and ≥five CTCs per 7.5 mL of blood. A cut-off of both ≥two and ≥five CTCs per 7.5 mL pf blood predicts a significantly shorter overall survival time for patients with CTCs (*p* = 0.027 and *p* = 0.008, respectively, log-rank test) (Figure 2A,B). We further performed survival analysis on different subgroups. We next analyzed all of the brain metastasis patients, regardless of having further overt metastases. Again, we found a significantly shorter overall survival when CTCs were detected than for CTC-negative patients, for both cut-offs ≥two and ≥five (*p* = 0.002 and *p* = 0.001, respectively). This underline that although infrequently detected, CTC detection in patients with brain metastases is of prognostic significance (Figure 2C,D). No survival analysis could be performed with a cut-off of ≥two CTCs in oligo-brain metastasis patients, as only two patients were CTC-positive. However, using ≥one CTC as a cut-off, we observed a significantly shorter overall survival time in the case with ≥one CTC (*p* = 0.019, log-rank test) (Figure 2E). Remarkably, all of the oligo-brain CTC-positive patients died between 1.3–8.0 months of follow-up, with a mean overall survival time of 4.6 months. In contrast, CTC-negative oligo-brain metastatic patients had a mean overall survival of 10.4 months (range: 0.7–47.7). When analyzing the overall survival of patients with lung metastases, we observed a significantly shorter overall survival time when patients were identified as CTC-positive for both CTC cut-off values (*p* = 0.004 and *p* < 0.0001, log-rank test) (Figure 2F,G), whereas no survival effect was observed for patients with bone or adrenal gland metastasis.

Cox proportional hazard function identified both CTC cut-offs as independent prognostic factors for the whole study cohort (two ≥ CTC HR: 1.629, *p* = 0.024, 95% CI: 1.137–6.465 and five ≥ CTC HR: 2.846, *p* = 0.0304 95% CI: 1.104–7.339), as well as for the brain metastases patients (two≥ CTC HR: 4.694, *p* = 0.004, 95% CI: 1.650–13.354 and five≥ CTC HR: 4.963 *p* = 0.003, 95% CI: 1.752–14.061). For patients with lung metastases only, the cut-off of five≥ CTCs was significant (five≥ CTC HR: 36.790, *p* = 0.009, 95% CI: 2.455–551.351) (Appendix A). Therefore, we concluded that CTCs could serve as prognostic markers in NSCLC patients in general, and especially for those with brain metastases.

### 2.5. CTC Detection in Brain Metastatic Patients of Mixed Primary Tumor Entities

To analyze whether the rate of CTC positivity in NSCLC cancer patients with brain metastasis is comparable to that of patients suffering from other tumors with brain metastasis, peripheral blood was collected from 52 cancer patients of different tumor entities prior to the surgical resection of brain metastasis. We recruited blood samples from patients with colorectal carcinomas (*n* = 22) and 30 patients with other epithelial tumor entities, including small-cell lung cancer (SCLC), prostate, and ovarian carcinomas. Since the accepted CTC cut-off value for patients with metastatic colon cancer is ≥three CTCs, samples with ≥three CTCs per 7.5 mL of blood were defined as CTC-positive in this analysis. In total, at the time of metastatic resection, 11.1% (one out of nine) of SCLC, 4.5% (one out of 22) of colon cancer, and 14.3% (three out of 21) of patients in the mixed group were identified as CTC-positive (Figure 3, Table 3). Therefore, we concluded that brain metastatic patients harbor remarkably few CTCs independent from their primary tumor histology. Thus, CTC rarity is a brain-specific rather than an NSCLC-specific effect.

## 3. Discussion

In this study we analyzed the presence of CTCs in a cohort of NSCLC patients with a particular focus on patients suffering from brain metastases. Furthermore, we analyzed CTCs in a cohort of patients with brain metastases from other tumor entities, including colon, SCLC, ovarian, and renal cell carcinomas. We could demonstrate that brain metastatic patients harbor a reduced frequency of CTCs compared to patients with other metastases, irrespective of their primary tumor origin. Interestingly, although EpCAM-positive CTCs were only found in a small proportion among brain metastatic patients, they were of high clinical relevance with regard to their prognostic significance. In addition, CTC-positive oligo-metastatic patients had a particularly bad prognosis, underlining a potential role of CTC analysis in the decision-making process of treating oligo-metastatic patients.

Our reported CTC positivity rate of 13.3% (≥two CTCs/7.5mL of blood) among metastatic NSCLC patients was low compared to other studies using the CellSearch system (15–37% CTC positivity using a cut-off level of ≥two to five CTCs/7.5mL of blood) [5,6,23,24]. We identified less CTC-positive patients, which was probably due to the large proportion of patients with brain as the single metastatic site in our cohort, with only 5.9% of them being CTC-positive (≥two CTC). In contrast, we observed a CTC positivity rate of 42.9% and 26.1% (≥three CTCs), respectively, in patients with adrenal gland or lung metastases, which is in line with the results presented in other reports [25,26]. These results indicate a strong impact of the metastatic site and number of metastases/tumor load on CTC frequencies. Also, the recirculation of CTCs from distant metastatic sites could be of importance here. However, hypothetically, the recirculation of CTCs seems to be more efficient from sites other than the brain.

To determine whether the low CTC frequency in lung cancer brain metastasis patients is a unique feature for lung cancer, peripheral blood from 52 other cancer patients of different tumor entities was collected prior to the resection of brain metastasis. This data confirmed the low CTC numbers among all of the patients with brain metastases (9.6%, ≥three CTC/7.5 mL of blood). The CellSearch system is cleared by the American food and drug administration (FDA) for metastatic colon cancer, with reported positivity rates of approximately 30–40% (≥ three CTCs/7.5mL of blood) [27]. In our brain metastasis cohort, only 4.6% of colon cancer patients had CTCs. Similarly, SCLC patients have been reported to display very high CTC counts [28]. In contrast, we only had one brain metastatic SCLC patient with ≥ three detectable CTCs (11.1%).

Our findings point towards a possibly diverse biological pattern of CTC dissemination in patients with brain metastases. Remarkably, although glioblastoma is an aggressive and highly invasive brain tumor, it rarely colonizes extracranial organs. However, about 20% of glioblastoma patients have been detected as CTC-positive in peripheral blood [29,30]. The question is whether the immune system prevents the colonization of glioblastoma-derived CTCs outside the brain, while being rather inefficient inside the brain [25]. Similarly, tumor cells from brain metastases might have adapted to the brain environment, so that an altered CTC phenotype is detected by the immune cells directly after disseminating from the brain. Alternatively, another hypothesis could be that the blood–brain barrier, especially among the oligo-metastatic cases, could still be rather intact, and thus could prevent the recirculation from the brain to the peripheral blood. A third scenario could be that patients with brain metastases have a higher proportion of CTCs that have undergone epithelial-to-mesenchymal transition (EMT), and thus are EpCAM-negative, and remain undetectable by the CellSearch system. Interestingly, also in breast cancer, fewer CTCs have been reported for brain metastatic patients when analyzed by the CellSearch system [26], and those CTCs that were competent to colonize the brain as a distant target organ site were predominantly EpCAM-negative but EGFR, NOTCH1 and HPSE-positive [31]. Importantly, these EpCAM-negative CTCs had a higher expression of CD44, a lower expression of CDH1, and a generally higher EMT signature [32]. We recently reported that different subpopulations of CTCs can be detected in advanced NSCLC patients, and that the CTC frequency was much higher when CTCs were detected by EMT/stem-cell related marker expression [33]. However, the clinical relevance of these different subpopulations of CTCs remains unclear. Clearly, additional studies using different CTC isolation techniques need to be performed to clarify whether the low CTC yields in this study are caused by the patient population or by choice of the detection method.

Interestingly, although fewer CTCs were found among brain metastatic patients, we demonstrated a strong association with patient survival, indicating that EpCAM-positive cells are of clinical relevance, irrespective of the potential existence of other EpCAM-negative CTCs. Furthermore, our data suggests that the presence of EpCAM-positive CTCs is associated with the number of distant metastases. In our study, patients with only brain metastases (oligo-metastatic disease) harbor fewer CTCs compared to patients in a multi-metastatic disease setting. A recent meta-analysis revealed that oligo-metastatic patients have a significantly increased five-year overall survival compared to multi-metastatic patients, and thus, an oligo-metastatic disease status might represent a distinct patient group among metastatic NSCLC [22]. Here, a single CTC among oligo-metastatic patients was associated with a significantly shorter overall survival, with a median survival of only 3.5 months. Thus, CTC-positive patients within the oligo-metastatic subgroup were at risk for a poor prognosis, and should be considered for aggressive treatment options. However, this result is based on low total patient numbers, which is a clear limitation of this study. In order to draw a strong clinical conclusion, this result needs to be confirmed by a larger study, including patients with other types of oligo-metastatic NSCLC.

EpCAM has been shown to be a marker for human embryonic stem cells, and thus has been used in several studies as a marker for pluripotency [34]. EpCAM has also been implicated as a marker for cancer stem cells (CSC) [35]. Al-Hajj et al. showed that the EpCAM, CD44, and CD24-positive CSCs had a >10-fold higher frequency of tumor-initiating capability than the EpCAM-negative cells [36]. Similarly, Baccelli et al. showed that CTCs that were derived from human breast cancer patients and were competent enough to give rise to bone, lung, and liver metastases in mice were positive for EpCAM, CD44, CD47, and MET [37]. Therefore, also in brain metastatic NSCLC patients, the detected EpCAM-positive CTCs could indicate CSCs properties, and would thus explain the highly significant association of even one CTC/7.5 ml with a poor patient outcome.

Taken together, although we did not observe a significantly longer survival in oligo-metastatic brain patients compared to that of multi-metastatic patients, we demonstrate that CTC detection can serve as a prognostic factor for overall survival in patients with isolated metastatic dissemination to the brain. Furthermore, our data on CTCs supports the idea that oligo-metastasis is a disease state between a primary tumor setting and a fully blown metastatic setting. CTC analyses could hereby serve as a biomarker to select patients who can profit from a radical treatment of their primary tumors and metastasis. In any case, these interesting observations would need to be elucidated in more clinical depth, in a larger patient cohort, and also with the support of EpCAM-independent isolation and detection techniques.

## 4. Materials and Methods

### 4.1. Patient Cohort

To analyze CTC frequencies in peripheral blood from metastatic patients with an oligo-brain or a systemic metastatic disease setting, a total number of 142 advanced stage cancer patients were enrolled at the University Medical Center Hamburg Eppendorf (UKE, Hamburg, Germany) between June 2013 and May 2017. Patients were either treated at the Department of Neurosurgery or at the Department of Internal Medicine at the UKE, and blood was collected either prior to the brain metastases surgery or before the start of a new line of treatment (chemotherapy, radiotherapy, or targeted therapy). Altogether, blood was collected from 36 NSCLC patients during treatment and from 52 patients before any type of treatment. Pre-treatment data was missing from two patients. Brain metastasis was assessed by Magnetic Resonance Imaging (MRI) and performed with gadolinium as the contrast-enhancing agent. Patients with leptomeningeal metastases were excluded from the study. The study was carried out in accordance with the World Medical Association Declaration of Helsinki and the guidelines for experimentation with humans by the Chambers of Physicians of the State of Hamburg. The experimental protocol was approved (Approval No. PVN-3779) by the Ethics Committee of the Chambers of Physicians (Berlin, Germany). All of the participants provided written informed consent. In total, 90 NSCLC patients—51 male and 39 female patients—were recruited. In 10 patients, an activating EGFR mutation was detected, whereas in 59 patients, no mutation was found (not assessed in 29 patients). None of the patients were diagnosed to carry an ALK translocation. In total, 56 NSCLC patients were suffering from brain metastases, with 34 having the brain as a single metastatic site (oligo-brain), and 22 having additional distant metastases. Twenty-six of the NSCLC cases had singular brain metastases, whereas 28 cases had ≥two metastases in the brain (average 2.3 metastases, data on number of metastases was missing for two cases). In 31 NSCLC patients, the blood was collected before the brain metastasis resection. Out of these patients, 25.8% had also extracranial metastases at the time point of brain operation. Further detailed patient characteristics are reported in Table 1. In addition, blood from nine small-cell lung cancer or large-cell neuroendocrine lung cancer, as well as from 43 patients harboring other tumor entities with brain metastases, were also collected. A total of 22 samples originated from patients with colon cancer, four originated from patients with renal cell carcinomas, five originated from patients with ovarian cancers, seven originated from patients with cancers of unknown primary causes (CUP), and five originated from patients with mixed tumors. Tumor histology and patient gender are listed in Table 2.

### 4.2. CTC Isolation and Detection by CellSearch System

For the CellSearch^®^ analysis, 7.5 mL of blood were drawn into CellSave tubes (Immunicon, Inc., Huntingdon Valley, PA, USA) and processed within 72 hours. The automated CTC analysis was performed as described above [38]. EpCAM-coated magnetic particles were used to isolate CTC, which were then identified via a pan-keratin and DAPI staining in an automated scanning process. CD45 immunofluorescence staining was used as a leukocyte marker. Cells were evaluated by an experienced scientist (SR). In 15 CTC-positive NSCLC patients, EGFR expression in CTC was additionally assessed using the fourth fluorescence channel of the CellSearch System, and expression intensity was assessed as described before [39].

### 4.3. Statistical Analysis

The CellSearch System (Menarini Silicon Biosystems, Huntington Valley, PA, USA) has been cleared by the FDA for CTC analyses in metastatic breast, colorectal, and prostate cancer. For colorectal cancer, a cut-off of ≥three CTCs was set [25], whereas for metastatic breast and prostate cancer patients, a cut-off of ≥five CTCs was defined [40,41]. In NSCLC, less CTCs were detected when using the CellSearch System [6]. A recent pooled analysis of CTCs in advanced NSCLC confirmed CTCs as an independent prognostic indicator of progression-free survival and overall survival in late stage NSCLC when using either ≥two CTCs or ≥five CTCs. CTC count also improved the prognostication when added to full clinicopathological predictive models [42]. Therefore, we employed also cut-offs of both ≥two CTCs and ≥five CTCs for the analyses.

The G-test of independence with Williams’ correction was used to identify group differences and associations between investigated variables and clinicohistopathological risk factors using In-Silico Online version 2.0 (Hamburg, Germany) [43]. A two-sided *p*-value ≤ 0.05 was considered statistically significant. Kaplan–Meier estimates in combination with the log-rank test were used to analyze survival differences between the groups.

## 5. Conclusions

Our data indicates that although patients with brain metastases more seldom harbor CTCs, they are still predictive for overall survival, and CTCs might be a useful biomarker to identify oligo-metastatic NSCLC patients who might benefit from a more intense therapy.

## Figures and Tables

**Figure 1 cancers-10-00527-f001:**
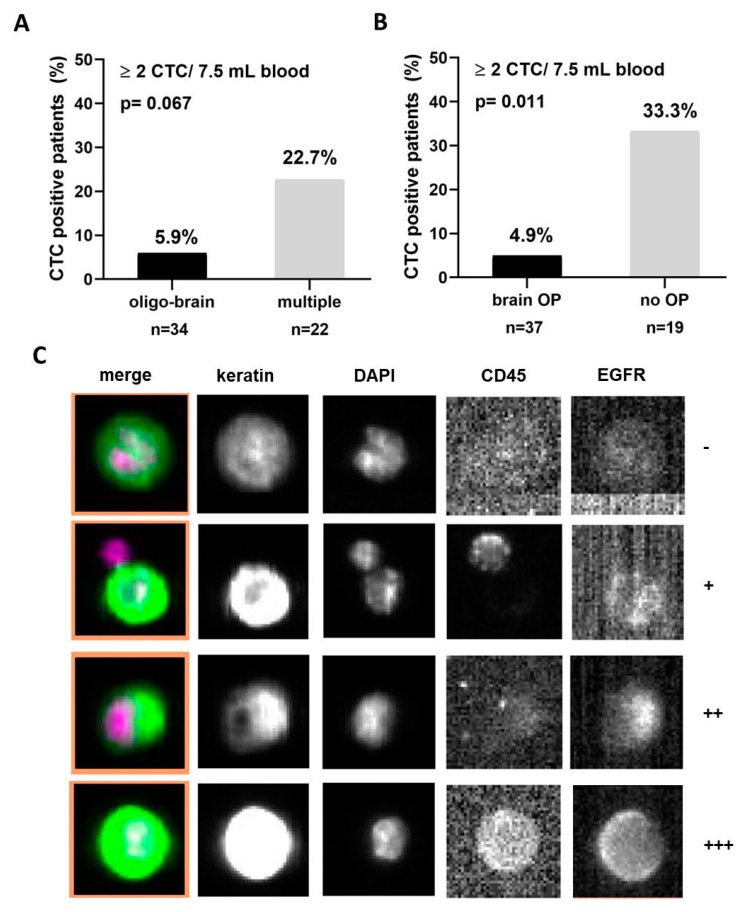
Circulating tumor cells (CTC) detection in brain metastasis non-small cell lung cancer (NSCLC) patients: (**A**) NSCLC patients with an oligo-brain disease are less frequently identified as CTC-positive compared to patients with multiple metastatic sites, including brain. (**B**) NSCLC patients that were treated by brain metastasis resection harbor CTCs significantly less frequently compared to brain metastasis patients treated by radiation or chemotherapy. (**C**) Representative pictures of EGFR-positive and EGFR-negative CTCs detected in an NSCLC patient. −: no EGFR expression; +: weak; ++: intermediate; +++: strong EGFR expression. OP: operation.

**Figure 2 cancers-10-00527-f002:**
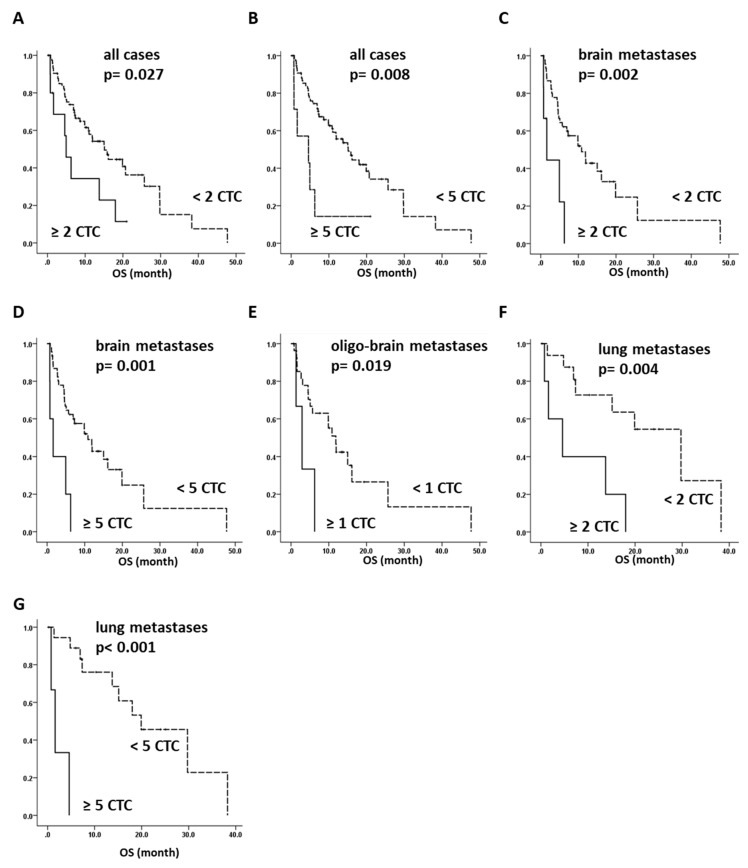
NSCLC patient overall survival analysis: Overall survival of metastatic NSCLC is significantly decreased when either two (**A**) or five (**B**) CTCs were detected. Brain metastases patients have a significantly decreased overall survival time when positive for either ≥two or ≥five CTCs (**C**,**D**). Overall survival of oligo-brain metastasis patients is significantly decreased upon the presence of CTCs (≥one) (**E**). CTC positivity is significantly associated with a shorter overall survival also in patients with lung metastases (≥two and ≥five CTCs) (**F**,**G**).

**Figure 3 cancers-10-00527-f003:**
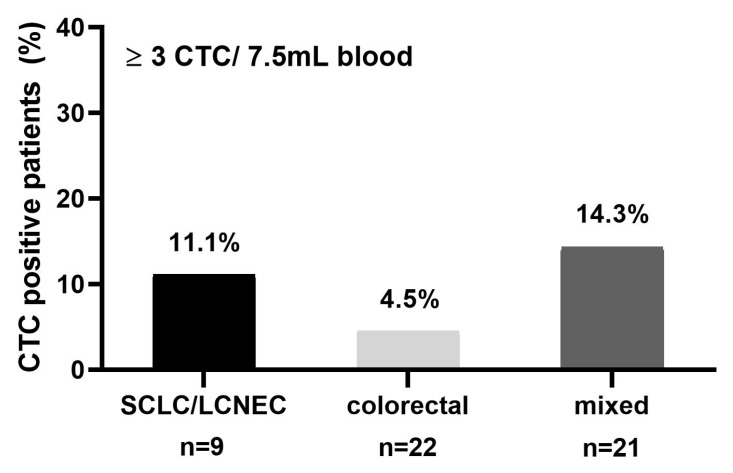
CTC detection in brain metastatic patients of mixed primary tumor entities: CTC detection at the time of brain metastasis resection from small-cell lung cancer (SCLC) and large-cell neuroendocrine cancer (LCNEC), colorectal cancer, and other epithelial tumors.

**Table 1 cancers-10-00527-t001:** Clinical characteristics of non-small cell lung cancer (NSCLC) patients, circulating tumor cells (CTC) status, and statistical association.

Clinical Characteristics		*n*	%	Negative ≤1 CTC	Positive ≥1 CTC	*p*-Value	Negative ≤3 CTC	Positive ≥3 CTC	*p*-Value
Gender	Female	46	54.1	82.6	17.4	0.429	91.3	8.7	1
	Male	39	45.9	74.4	25.6		89.7	10.3	
Histology	AC	61	76.3	75.4	24.6	0.539	88.5	4.9	0.672
	SCC	19	23.8	84.2	15.8		94.7	5.3	
	n.a.	5							
Brain Metastasis	Yes	51	60.0	80.4	19.6	0.788	88.2	11.8	0.467
	No	34	40.0	76.5	23.5		94.1	5.9	
Brain Operation	Yes	41	80.4	87.8	12.2	0.017	95.1	4.9	0.01
	No	10	19.6	50.0	50.0		60.0	40.0	
Oligo-Brain	Yes	30	60.0	86.2	13.8	0.279	96.7	3.3	0.035
	No	20	40.0	70.0	30.0		75.0	25.0	
	n.a.	1							
Bone Metastasis	Yes	15	17.6	60.0	40.0	0.077	80.0	20.0	0.144
	No	70	82.4	82.9	17.1		82.9	7.1	
	n.a.								
Adrenal Gland Metastasis	Yes	14	16.9	50.0	50.0	00.01	78.6	21.4	0.128
	No	69	83.1	84.1	15.9		92.8	7.2	
	n.a.	2							
Lung Metastasis	Yes	21	24.7	66.7	33.3	0.132	85.7	14.3	0.402
	No	64	75.3	82.8	17.2		92.2	7.8	

AC: adenocarcinoma SCC: squamous cell carcinoma; n.a.: data not available.

**Table 2 cancers-10-00527-t002:** EGFR expression in CTCs from NSCLC patients. Listed are the number of patients, the amount of detected CTCs, and the percentage of EGFR-expressing CTCs, with 1+ for weak, 2+ for intermediate, and 3+ for strong EGFR expression.

Patient	CTCs	EGFR+	Intensity
#1	7	28.60%	2+
#2	11	54.50%	1+
		27.30%	2+
		18.20%	3+
#3	7	42.90%	2+
#4	3	33.30%	1+
#5	11	36.40%	1+

**Table 3 cancers-10-00527-t003:** Gender, primary tumor origin, and CTC status (≥three CTC/7.5mL of blood) of brain metastasis patients recruited from different tumor entities.

Tumor	Samples	Female	Male	CTC neg.	CTC pos.
Type	n	n	n	%	%
All	52	27	27	90.4	9.6
SCLC & LCNEC	9	5	4	88.9	11.1
Colon	22	9	13	95.4	4.6
Renal	4	2	2	75.0	25.0
Ovarian	5	5	0	100.0	0.0
Mixed	5	1	4	80.0	20.0
CUP	7	5	4	85.7	14.3

SCLC: small cell lung cancer; LCNEC: large-cell neuroendocrine tumor; CUP: cancer of unknown primary.

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
