# Peer review of "Frequency of Circulating Tumor Cells (CTC) in Patients with Brain Metastases: Implications as a Risk Assessment Marker in Oligo-Metastatic Disease"

_cancers, 2018, doi:10.3390/cancers10120527_

Round 1

Reviewer 1 Report

(1) CTCs in NSCLC patients are difficult to study due to their rareness. The main weakness of the present work is the lack of data on patient treatment. Metastatic NSCLC is treated either with TKIs/immunotherapy or chemotherapy and the kind of care will critically influence the presence of CTCs. In particular, some TKIs prevent or cure brain metastases and, therefore, the detection of wildtype-EGFR metastases in the present study is not surprising. Data on the tratment of the patients and the time point of blood collection for CTC analysis need to be supplemented.

Using the CellSearch® system, few CTCs were detected among NSCLC patients with brain metastases (n=52, 12.5%, ≥2 and 8.9% ≥5 CTC/7.5ml blood) and especially oligo-brain metastatic patients (n=34, 5.9% and 2.9%).

(2). There are reports demonstrating that brain metastasis CTC are EpCam-negative. This discussion should be included in more detail. For example: CTCs isolated from PBMCs of patients with breast cancer [doi: 10.1126/scitranslmed.3005109]. In epithelial cell adhesion molecule (EpCAM)–negative CTCs, we identified a potential signature of brain metastasis comprising “brain metastasis selected markers (BMSM)” HER2+/EGFR+/HPSE+/Notch1+. These CTC are not captured by the CellSearch platform because of their EpCAM negativity.

Our data indicates that although patients with brain metastases more seldom harbor CTCs, they are still predictive for overall survival, and CTCs might be a useful biomarker to identify oligo-metastatic NSCLC patients who might benefit from a more intense therapy.

(3) CTC are not only produced by primary tumors but likewise are contributed by metastases – thus, a lower number of CTCs in brain-only metastatic patients seems not very surprising. This point needs to be discussed.

Prognostic indicators for shorter overall survival time among all 24 NSCLC patients (n=90, 2≥ HR:1.629, p=0.024, 95% CI:1.137-6.465 and 5≥ HR:2.846, p=0.0304, CI:1.104- 25 7.339) and among patients with brain metastases (2≥ HR:4.694, p=0.004, CI:1.650-13.354 and 5≥ 26 HR:4.963 p=0.003, CI:1.752-14.061). Also oligo-brain NSCLC metastatic patients with CTCs had a 27 very poor prognosis (p=0.019). Similarly, in other tumor entities only 9.6% of the patients with brain 28 metastases (n=52) had detectable CTCs.

(4) The conclusions for a prognostic validity of the CTCs of brain metastases-positive patients are based on 2 CTCs and 12 cases, which needs confirmation by more and larger studies, preferentially supplemented by methods independently of EpCam expression (should be included).

(5) Supporting evidence is drawn from CTC analysis of mixed primary tumor entities, including colon cancer, SCLC and others. This is not convincing; for example, most of metastatic SCLC patients were shown to carry high numbers of CTCs (Dive group) and brain metastasis is tried to be prevented by prophylactical cranial irradiation.

(6) In conclusion, this work describes an interesting observation but the conclusions are weakly supported by the data due to methodology and the lack of description of clinical care supplied which greatly modulates CTC numbers. At least, all these limitations should be clearly stated in a revised paper.

Author Response

Review 1

1) CTCs in NSCLC patients are difficult to study due to their rareness. The main weakness of the present work is the lack of data on patient treatment. Metastatic NSCLC is treated either with TKIs/immunotherapy or chemotherapy and the kind of care will critically influence the presence of CTCs. In particular, some TKIs prevent or cure brain metastases and, therefore, the detection of wildtype-EGFR metastases in the present study is not surprising. Data on the tratment of the patients and the time point of blood collection for CTC analysis need to be supplemented.

The reviewers raise an important point. Indeed, if the blood would have been collected from patients during a remission phase of the disease clearly less CTCs would be anticipated. We have now clarified this issue by including a description of patient pre-treatment into materials & methods. The blood was collected from patients either at first diagnosis (i.e. patients without prior treatment), or during treatment at relapse. Altogether blood was thus collected from 36 patients during progressive disease/ relapse and thus during treatment and from 52 before any type of treatment. From 2 patients the data of pre-treatment is missing. Therefore, we believe that the low CTC number is not related to patient treatment. We have now deleted the TNM data from table 1. As the TNM status only gets evaluated at initial diagnosis and did therefore not necessarily reflect the clinical staging at the time point of blood withdrawal, it could have been misleading. In our cohort only 10 patients were carriers of an EGFR mutation. From 5 of these patients the blood was collected at time of diagnosis and from the other five at the time of progressive disease during EGFR TKI treatment. Therefore, our study design is clearly not suitable to answer the question if the rather low EGFR expression on CTCs in NSCLC is caused by a a priori TKI treatment.  

(2). There are reports demonstrating that brain metastasis CTC are EpCam-negative. This discussion should be included in more detail. For example: CTCs isolated from PBMCs of patients with breast cancer [doi: 10.1126/scitranslmed.3005109]. In epithelial cell adhesion molecule (EpCAM)–negative CTCs, we identified a potential signature of brain metastasis comprising “brain metastasis selected markers (BMSM)” HER2+/EGFR+/HPSE+/Notch1+. These CTC are not captured by the CellSearch platform because of their EpCAM negativity.

We have now extended the discussion of EpCam negative CTCs and have added also other breast cancer references supporting the fact that breast cancer brain metastases may have a large proportion of EpCam negative CTCs.  However, to our knowledge this is the first papers describing the CTCs phenotypes in NSCLC brain metastases. The CellSearch system is known to detect intermediate CTCs with low EpCam expression and low keratin expression, often defined as intermediate EMT CTCs. Intermediate CTCs have been suggested to be the most aggressive cells. This may explain the strong prognostic power of all the CellSearch results published including our results.

(3) CTC are not only produced by primary tumors but likewise are contributed by metastases – thus, a lower number of CTCs in brain-only metastatic patients seems not very surprising. This point needs to be discussed.

We agree that CTCs are most likely shed by both primary tumors and metastases. Therefore, CTC counts are probably related to the total tumor load but more importantly to the disease state of the patient (see first point). The clinical data implies that the oligo- metastatic state is a disease state between the primary tumor setting and a fully blown metastatic setting. Thus, the low CTC amounts are not that surprising but, most importantly, albeit the low frequency of high prognostic relevance. We have now included this into the discussion. However, please note that the number of brain mets did not correlate with CTC amounts. In 15 out of the 33 (1 missing data) oligo- brain cases patients had more than one brain met (with up to 5 mets). This supports that patients with brain mets indeed have generally less EpCam-positive CTCs.

(4) The conclusions for a prognostic validity of the CTCs of brain metastases-positive patients are based on 2 CTCs and 12 cases, which needs confirmation by more and larger studies, preferentially supplemented by methods independently of EpCam expression (should be included).

We totally agree that the numbers are rather low and needs to be confirmed by other independent studies, preferentially using other independent methods. We have now added this to the discussion.

(5) Supporting evidence is drawn from CTC analysis of mixed primary tumor entities, including colon cancer, SCLC and others. This is not convincing; for example, most of metastatic SCLC patients were shown to carry high numbers of CTCs (Dive group) and brain metastasis is tried to be prevented by prophylactical cranial irradiation.

We are sorry but we don’t understand the question. All our CellSearch analysis were performed blinded (regarding clinical data) by experienced scientists (double calling by SR and HW). We have been involved in several clinical trials involving CellSearch and thus we believe that our CellSearch data is of high quality. The results obtained are those shown in the paper and are representative for patients undergoing brain surgery. Clearly again here the tumor disease state of the patients probably plays an important role and the results are not comparable to patients with end stage patients. Only patients in rather good shape can be operated. Thus our SCLC patients are rather untypical patients as these patients are seldom operated. Of the 11 patient 4 were operated as CUP (cancer of unknown primary), only later their SCLC diagnosis was confirmed.

(6) In conclusion, this work describes an interesting observation but the conclusions are weakly supported by the data due to methodology and the lack of description of clinical care supplied which greatly modulates CTC numbers. At least, all these limitations should be clearly stated in a revised paper.

We have now based on the good points raised above clearly stated the limitations of our paper.

Reviewer 2 Report

Reviewers' Comments to Author:

In this study design, author demonstrated the significance of CTC measurements showed the predictive values for overall survival, and CTCs might be a useful biomarker to identify oligo-metastatic NSCLC. The manuscript was well-written especially about altered CTC phenotype with brain metastasis in discussion part. Author should clarify the importance of measurement CTC in patients with oligo-metastatic brain disease for precision medicine. Furthermore, it seems quite difficult to compare in a cross-sectional way among the different kinds of tumor types with brain metastasis and reach to conclude the brain metastatic patients harbor remarkably few CTCs independent from their primary tumor histology without same solid patients with comparing multi-brain or other distal organs metastasis group and with different cut-off CTC number compared to NSCLC. This information provide us further confusion and not reliable and thus may be removed.

Major comments:

If all brain metastasis analyzation showed statistically difference in OS between ≥ 1 and < 1 CTC group in Fig 2. Is there any meaning of patients selective measurement CTC in patients with oligo-metastatic brain disease? Or its p-value is better than multi-metastatic brain disease analysed by cut-off CTC number as ≥ 1 or < 1? How about in patients with multi-metastatic brain disease (n=22)? Author should address and emphasize about clinical benefit for measurement of CTCs in patients with oligo-metastatic brain in discussion.

The 2/32 oligo-brain patients showed more CTCs (5.9%, ≥ 2). Why these two oligo-brain patients showed more CTCs?  Rapidly recurrence? Their primary tumor showed continuous spread? High CTCs number with oligo-metastatic brain disease showed poor OS may suggest still exist micro metastatic lesion in other brain or other distant sites. How difference clinical time course of these poor prognostic patients. Did you follow up and actually find the rapid recurrences of metastatic lesion by following up CT? Recurred at other distal organ destroyed by BBB or spreading oligo-brain metastatic lesion?

Is there any difference in %CTC of oligo-other distal organ metastatic patients, oligo- lung or bone or adrenal gland metastasis, compared to multi- metastasis? Only oligo-brain metastasis showed specific clinical outcome corresponding the number of detected CTCs? If so and author have enough sample number for analysis, author can emphasize more importance of the significance about the measurements of CTC in oligo-brain metastatic patients.

Author should address more about the significance of EGFR expression on CTC. Did the EGFR expression ranged from weak to strong in CTC associate clinical outcome compared #1-3 to #4, 5? What`s kind of circulating cells which express EGFR positive but negative CTCs due to CD45 positive? Is this EPC? On the other hand, Keratin intensity seems positive correlation to EGFR intensity in other pictures. Author should address about this kind of CTC phenotype and the specific feature of both EpCAM and EGFR positive CTCs in discussion, although author mention only about EpCAM negative but EGFR, NOTCH1 and HPSE positive CTC, Ref31.

The detail clinical information of other tumors with brain metastasis is not clarified. Were all of patients with other tumor also oligo-metastatic brain disease? Why the author choose different cut-off number less/more 3 CTCs and different from present NSCLC? If author select cut-off number as 3 CTCs due to the accepted CTC cut-off value for patients with metastatic colon cancer. Author cannot compare in a cross-sectional way among the different kinds of tumor types and reach to conclude the brain metastatic patients harbor remarkably few CTCs independent from their primary tumor histology without comparing brain metastasis group. And furthermore, sample size of each tumor is small and it`s hard to say by mix together of different organ origin heterogeneous tumors. Although author refer 25 and 26 to proof low CTC numbers among all patients with brain metastases for comparison, this information provide us further confusion and is not reliable and thus may be removed.

Miner comment:

Merged pictures shows quite confusion DAPI shows magenta? What fluorescence colors about EGFR and keratin? Why center of color in merge CTC picture showed white, but not magenta according to the location of DAPI, although, magenta color located in nuclear area of CTC corresponding DAPI location. Is this due to over halation? No labeling C in Fig 1. What is meaning brain OP in Fig 1B, operated?

Author Response

Review  2

In this study design, author demonstrated the significance of CTC measurements showed the predictive values for overall survival, and CTCs might be a useful biomarker to identify oligo-metastatic NSCLC. The manuscript was well-written especially about altered CTC phenotype with brain metastasis in discussion part. Author should clarify the importance of measurement CTC in patients with oligo-metastatic brain disease for precision medicine. Furthermore, it seems quite difficult to compare in a cross-sectional way among the different kinds of tumor types with brain metastasis and reach to conclude the brain metastatic patients harbor remarkably few CTCs independent from their primary tumor histology without same solid patients with comparing multi-brain or other distal organs metastasis group and with different cut-off CTC number compared to NSCLC. This information provide us further confusion and not reliable and thus may be removed.

-          Thank you for your good comments. We have now added some clarifications regarding the importance of measurement CTC in patients with oligo-metastatic brain disease. The discovery of potent biomarkers to stratify patients with cerebral metastasis into a high and a low risk group potentially enables the identification of patients who might benefit from aggressive therapy and extended adjuvant treatment regimens. For example, if a solitary brain metastasis has been resected without radiographic evidence of peripheral filiae, the benefit or question of initiation of adjuvant systemic chemotherapy is not clear. Given the increasing efficacy of systemic targeted therapies and the rising numbers of long-term survivors in many tumor entities, it is crucial to identify patients at risk and therapeutic windows to effectively induce tumor remission. Here, CTCs as peripheral biomarker can potentially identify patients with a higher risk for systemic disease progression. These high-risk patients, who display an active systemic tumor dissemination pattern, might benefit from aggressive systemic therapy. Our study shows that CTCs have the potential to serve as biomarker in patients with oligo-metastatic brain disease.

-           In this study we chose to analyze the CTCs using the FDA approved CellSearch system (for breast, colon and prostate as stated in materials and methods). The rationale to choose this system is that data from many 10 000 patients have been collected so far. In our laboratory we have had the device for over 10 years and used the machine in several clinical trials. Thanks to the standardized analysis mode the data from different studies can be easily compared to each other. We therefore feel comfortable to state that “brain metastatic patients harbor remarkably few CTCs independent from their primary tumor histology” as our brain metastatic patients had clearly less CTCs than what many large single institution studies and meta-analyses have reported. The rational to choose different CTC cut offs is also based on the FDA recommendations. Different cut offs are used for different tumor entities. Therefore, we decided to choose an average cut-off of 3 CTCs based on the official cut-off for colon. For NSCLC however, we used the cut-off of ≥2 and ≥5 CTCs as for this entity a meta-analysis showed that these cut offs have the highest clinical relevance. We therefore believe the information is of general interest and we would like to keep the data.

Major comments:

1.       If all brain metastasis analyzation showed statistically difference in OS between ≥ 1 and < 1 CTC group in Fig 2. Is there any meaning of patients selective measurement CTC in patients with oligo-metastatic brain disease? Or its p-value is better than multi-metastatic brain disease analysed by cut-off CTC number as ≥ 1 or < 1? How about in patients with multi-metastatic brain disease (n=22)? Author should address and emphasize about clinical benefit for measurement of CTCs in patients with oligo-metastatic brain in discussion.

Based on the NSCLC meta-analysis data mentioned above we tried to use the same cut-off for all NSCLC samples. However, due to the rareness of CTCs in oligo met patients this was not possible. Therefore, we used a different cut-off particularly for this group. This is also done within other tumor entities. For example, in non-metastatic breast cancer patients - a different cut-off is used compared to the metastatic setting (1 CTC versus 5 CTCs). Their predictive power has been shown in many large meta-analysis (Janni et al., Clinical Cancer res 2016, Bidard et al., Lance Oncology 2014).  Interestingly, a cut-off of 1 CTC was not statistically significant for the whole NSCLC study group. Therefore, we think that also our data on CTCs supports the idea that the oligo-metastatic setting is a disease state between a primary tumor setting and a fully blown metastatic disease. Current diagnostic protocols favor the radiographic staging via CT of the thorax and abdomen. This diagnostic approach, however, can only detect established tumors >0.5-25px in size. Dormant or rarely disseminated cells cannot be detected by CT scans. On the other hand CTC screening can identify metastatic spread during the early stages. Physicians can therefore detect the active metastatic dissemination earlier and treat accordingly. Especially, since CTCs can be analyzed on a single cell level and give valuable information about the molecular tumor phenotype. This issue has now been discussed in more detail including the benefit of measurement of CTCs in patients with oligo-metastatic brain disease.

2.       The 2/32 oligo-brain patients showed more CTCs (5.9%, ≥ 2). Why these two oligo-brain patients showed more CTCs?  Rapidly recurrence? Their primary tumor showed continuous spread? High CTCs number with oligo-metastatic brain disease showed poor OS may suggest still exist micro metastatic lesion in other brain or other distant sites. How difference clinical time course of these poor prognostic patients. Did you follow up and actually find the rapid recurrences of metastatic lesion by following up CT? Recurred at other distal organ destroyed by BBB or spreading oligo-brain metastatic lesion?

One of these patients had a solitary brain met that could not be operated and received only irradiation therapy combined with chemo therapy after the diagnosis (and blood drawl). The primary tumor was not either operated and the patient died 6 months after the diagnosis. The second patient was operated for a brain met recurrence. The first brain met was operated one year before the recurrence. The patient was not treated at the hospital after the operation and thus no follow up is available for this patient. Thus both patients had a progredient oligo-metastatic disease at the time of blood collection.

3.       Is there any difference in %CTC of oligo-other distal organ metastatic patients, oligo- lung or bone or adrenal gland metastasis, compared to multi- metastasis? Only oligo-brain metastasis showed specific clinical outcome corresponding the number of detected CTCs? If so and author have enough sample number for analysis, author can emphasize more importance of the significance about the measurements of CTC in oligo-brain metastatic patients.

Thank you for this very good question. The sample numbers of the other groups are however very low and therefore thorough analyses of different subgroups cannot be performed. In the largest group of lung metastases (n=23) five patients had lung as only metastatic (relapse) site. Here, similar results were observed as we did in the brain, i.e. no CTCs were found among these patients. Clear conclusions based on this small patient number are hard to be drawn. Our data could only indicate that the higher the total tumor load the higher the probability of CTC detection, which goes hand in hand with the clinical notion of the better prognosis of oligo-metastatic patients in general. Therefore, our data may not be unique to brain, but due to lack of enough other oligo-met patients clear conclusions from other oligo-met situations cannot be drawn. We have now included this issue in the discussion stating that larger studies on different patient cohorts need to be performed.

Still importantly, the number of brain metastases did not correlate with CTC amounts. 15 out of the 33 (1 missing data) oligo-brain cases had more than 1 brain metastases (with up to 5 brain nodules), supporting the idea that patients having especially oligo-brain metastases clearly have in general less (EpCam-positive) CTCs.

4.       Author should address more about the significance of EGFR expression on CTC. Did the EGFR expression ranged from weak to strong in CTC associate clinical outcome compared #1-3 to #4, 5? What`s kind of circulating cells which express EGFR positive but negative CTCs due to CD45 positive? Is this EPC? On the other hand, Keratin intensity seems positive correlation to EGFR intensity in other pictures. Author should address about this kind of CTC phenotype and the specific feature of both EpCAM and EGFR positive CTCs in discussion, although author mention only about EpCAM negative but EGFR, NOTCH1 and HPSE positive CTC, Ref31.

The EGFR staining intensities & positivity rates in the CTCs between different patients did not correlate with any clinical data. This information we have now added into the results.

When using the CellSearch system the standard analysis software only shows potential CTCs that are based on Keratin, DAPI and CD45 staining. The fourth channel i.e. this time EGFR is not used to detect potential CTCs. Thus, EGFR is only an additional marker that can be assessed only on cells that were keratin and dapi positive and CD45 negative. Cells that are EGFR-positive but keratin negative are not recognized by the system and not shown by the software; neither are cells that are CD45 positive. Therefore, the EPCs cannot be analyzed using the standard analysis mode. In figure 1, four different CTCs are shown. Only by chance the EGFR negative CTC had a lower EGFR signal- no correlation between EGFR and keratin was generally observed. In conclusion, the role of EGFR alone could not be assessed using this platform.

5.       The detail clinical information of other tumors with brain metastasis is not clarified. Were all of patients with other tumor also oligo-metastatic brain disease? Why the author choose different cut-off number less/more 3 CTCs and different from present NSCLC? If author select cut-off number as 3 CTCs due to the accepted CTC cut-off value for patients with metastatic colon cancer. Author cannot compare in a cross-sectional way among the different kinds of tumor types and reach to conclude the brain metastatic patients harbor remarkably few CTCs independent from their primary tumor histology without comparing brain metastasis group. And furthermore, sample size of each tumor is small and it`s hard to say by mix together of different organ origin heterogeneous tumors. Although author refer 25 and 26 to proof low CTC numbers among all patients with brain metastases for comparison, this information provide us further confusion and is not reliable and thus may be removed.

Please see also previous answers. We have now analyzed the clinical data of the other metastases in more detail. Here we included only patients that could undergo a brain surgery. Only patients in rather good shape can be operated. Thus e.g. our SCLC patients are rather untypical patients as these patients are seldom operated. Of the 11 patient 4 were operated as CUP (cancer of unknown primary), only later their SCLC diagnosis was confirmed and five others had only brain metastases. Also in the colorectal-cancer (CCR) group eight persons had only brain metastases and naturally all CUPs have only brain metastases.  As mentioned above , the reason for using different cut offs is based on FDA recommendations and meta-analysis  data. CellSearch is cleared for breast prostate and CCR cancer. Different cut offs have been defined for different tumor entities.  For metastatic colorectal cancer a cut of off 3 CTCs is used and therefore all studies on CCR are using this cut off. As CellSearch is not cleared for NSCLC we used the cut off defined by the large meta-analysis. Here a cut off of wither 2 or 5 was shown to be of prognostic relevance. Based on the large very well comparable data already produced by the system on other metastases we feel comfortable to state that operable brain met patients harbor very few CTCs compared to other metastatic patients.  

Miner comment:

6.       Merged pictures shows quite confusion DAPI shows magenta? What fluorescence colors about EGFR and keratin? Why center of color in merge CTC picture showed white, but not magenta according to the location of DAPI, although, magenta color located in nuclear area of CTC corresponding DAPI location. Is this due to over halation? No labeling C in Fig 1. What is meaning brain OP in Fig 1B, operated?

The CTC images are produced by the CellSearch system and the images cannot be modified. The first panel shows the merged image in color with dapi shown in magenta and keratin in green. The white color is due to the high signal (saturated) of both dapi and keratin in the area. In the following B & W panels only the intensity of each fluorescence signal is shown.

           We are sorry for the missing labeling in Figure. We have now added this information,             i.e. brain OP, means patients undergoing brain surgery.

Round 2

Reviewer 1 Report

The authors have considered and answered the host of questions raised by both reviewers and this manuscript is now ready for publication in Cancers.

Reviewer 2 Report

I could agree with the significance of measurement of CTCs in patients with especially after resected oligo-metastatic brain disease for treatment decision as precision medicine by author response. I could understand the reason why author choose different cut-off among the different kind of cancers by using previous meta-analyses reports and the recommendation from FDA. Author further mention about the significance of EGFR-CTCs, current issues of present study design and future direction in discussion. Thus, it is already suitable for publication at this moment as acceptance.